# Structural role of essential light chains in the apicomplexan glideosome

Samuel Pazicky [1,2], Karthikeyan Dhamotharan [1,2], Karol Kaszuba[1,2], Haydyn D. T. Mertens [2], Tim Gilberger[1,3,4], Dmitri Svergun[2], Jan Kosinski[1,2,5], Ulrich Weininger[6] & Christian Löw [1,2 ✉]

Gliding, a type of motility based on an actin-myosin motor, is specific to apicomplexan parasites. Myosin A binds two light chains which further interact with glideosome associated proteins and assemble into the glideosome. The role of individual glideosome proteins is unclear due to the lack of structures of larger glideosome assemblies. Here, we investigate the role of essential light chains (ELCs) in *Toxoplasma gondii* and *Plasmodium falciparum* and present their crystal structures as part of trimeric sub-complexes. We show that although ELCs bind a conserved MyoA sequence, *P. falciparum* ELC adopts a distinct structure in the free and MyoA-bound state. We suggest that ELCs enhance MyoA performance by inducing secondary structure in MyoA and thus stiffen its lever arm. Structural and biophysical analysis reveals that calcium binding has no influence on the structure of ELCs. Our work represents a further step towards understanding the mechanism of gliding in *Apicomplexa*.

[1] Centre for Structural Systems Biology (CSSB), Notkestrasse 85, D-22607 Hamburg, Germany. [2] Molecular Biology Laboratory (EMBL), Hamburg Unit c/o Deutsches Elektronen Synchrotron (DESY), Notkestrasse 85, D-22607 Hamburg, Germany. [3] Bernhard Nocht Institute for Tropical Medicine, Bernhard-Nocht-Strasse 74, D-20359 Hamburg, Germany. [4] Department of Biology, University of Hamburg, Hamburg, Germany. [5] Structural and Computational Biology Unit, European Molecular Biology Laboratory, Meyerhofstrasse 1, 69117 Heidelberg, Germany. [6] Martin-Luther-University Halle-Wittenberg, Institute of Physics, Biophysics, D-06120 Halle (Saale), Germany. ✉email: christian.loew@embl-hamburg.de

Apicomplexa are a phylum of intracellular, parasitic, single cell eukaryotes with high medical and agricultural relevance. For instance, *Plasmodium species* are the causative agents of malaria, that lead to 414.000 deaths per year[1]. Another apicomplexan parasite, *Toxoplasma gondii*, infects more than 30% of the population worldwide with no clinical symptoms but can cause severe damage in immunocompromised patients and in pregnant women[2]. Proliferation and transmission of these obligate endoparasites in their host organisms rely on efficient cell invasion[3]. This active process is based on the motility of the parasite, referred to as gliding, and is empowered by an actin/myosin motor[4,5]. This motor is localized within the intermembrane space between the parasite's plasma membrane and inner membrane complex (IMC), an additional double-layer of membranes that is unique for these single cell organisms[6]. The IMC provides stability to invasion competent stages of the parasite and functions as an anchor for the actin/myosin motor. While motility is achieved by the interaction of the myosin motor with actin filaments, myosin is linked to the IMC by a membrane-embedded multi-protein complex referred to as the glideosome[7–9] (Fig. 1).

According to the current model, the apicomplexan glideosome is composed of six proteins: myosin MyoA, essential light chain ELC, myosin light chain MLC1, and the glideosome-associated proteins GAP40, GAP45 and GAP50[7,8,10]. MyoA is an unusually small myosin protein of the unconventional myosin class XIV[11,12], which lacks the typical myosin tail domain and binds the two light chains at the C-terminal myosin neck region[13,14]. MLC1 (in *P. falciparum*: myosin A tail-interacting protein, MTIP) binds at the very C-terminus of MyoA, while ELC is expected to interact with the C-terminus of MyoA upstream of MLC1[15]. Two ELC homologs recognizing the same MyoA region, termed TgELC1 and TgELC2, were identified in *T. gondii*[16], whereas only one PfELC homolog is known in *P. falciparum*[14,17].

Both light chains have been shown to stabilize MyoA in vivo and to be essential for parasite egress or invasion[16,18,19]. Myosin A and the light chains interact with the C-terminus of the glideosome associated protein 45 (GAP45) to form a pre-complex in the earlier stages of intracellular parasite development[7], which subsequently assembles with the remaining glideosome members (GAP40 and GAP50). N-terminal palmitoylation modification at its N-terminus anchors MLC1 (MTIP) to the IMC[20], whereas N-terminal myristoylation and palmitoylation sites tie GAP45 to the plasma membrane[10,21,22]. GAP45 is essential for the correct localization of MyoA with its light chains and GAP45 depletion leads to impairment of host cell invasion[10]. Depletion of GAP40 or GAP50 changes the morphology of the parasites and the integrity of the IMC and thereby also alters the localization of MyoA and the light chains[23].

Structural information on individual members and subcomplexes of the glideosome are limited and the architecture of the entire glideosome is elusive. So far, only structures of *P. falciparum* PfGAP50 soluble domain[24], a *T. gondii* dimeric complex between the TgMyoA C-terminus and MLC1[15], a homologous dimeric complex in *P. falciparum* between PfMyoA C-terminus and MTIP[25], and the motor domains of the *T. gondii* TgMyoA[26] and *P. falciparum* PfMyoA[27] are available (Supplementary Table 1).

Here, we present crystal structures of *T. gondii* and *P. falciparum* light chains bound to the respective MyoA C-termini in the presence of calcium, an additional calcium-free structure as well as the X-ray and NMR solution structures of the N-terminal domain of *P. falciparum* PfELC. We provide a thorough characterization of all identified interaction surfaces and discuss the differences between both species. We demonstrate that ELCs bind to a conserved binding site on MyoA to induce its α-helical secondary structure and stiffen the MyoA neck. Our work deepens the mechanistic understanding of the gliding motility in Apicomplexa.

## Results

**Structures of isolated ELCs.** Crystal structures of *T. gondii* and *P. falciparum* MyoA and of their distal light chains MLC1 (MTIP)[15,25] have already been determined. To shed light on the role of proximal essential light chains (ELCs), we studied their structure in isolation and in the context of their interaction partners. TgELC1 and TgELC2 share a high degree of sequence similarity (65.2%), whereas PfELC has only 40.6% similarity to TgELC1 (Supplementary Fig. 1a), pointing towards structural differences. Likewise, the disorder probability differs between *T. gondii* and *P. falciparum* ELCs (Supplementary Fig. 2a). We recombinantly expressed N-terminally His-tagged ELCs in *E. coli* (Supplementary Fig. 1b) and purified them to homogeneity. In spite of similar molecular weights, PfELC elutes earlier than TgELC2 when subjected to size exclusion chromatography (Supplementary Fig. 2b), indicative of a larger hydrodynamic radius for PfELC. Small angle X-ray scattering (SAXS) measurements further confirm that PfELC has a larger overall size in solution compared to TgELC2, with respective radii of gyration ($R_g$) of $2.71 \pm 0.05$ nm and $2.14 \pm 0.05$ nm (Supplementary Fig. 2d–e, Supplementary Table 2 and 3). The SAXS data also provide evidence that the increased $R_g$ of PfELC likely results from conformational flexibility (Supplementary Fig. 2f, Supplementary Table 3). This is also apparent from circular dichroism data which show that PfELC has lower α-helical and higher random coil content compared to TgELC2 (Supplementary Fig. 2c, Supplementary Table 2). To map the structured elements and disordered regions of PfELC, we performed triple-resonance NMR experiments that facilitated the near complete assignment of the amide backbone resonances (Supplementary Fig. 2j). Heteronuclear NOEs ({$^1$H}-$^{15}$N NOE) and

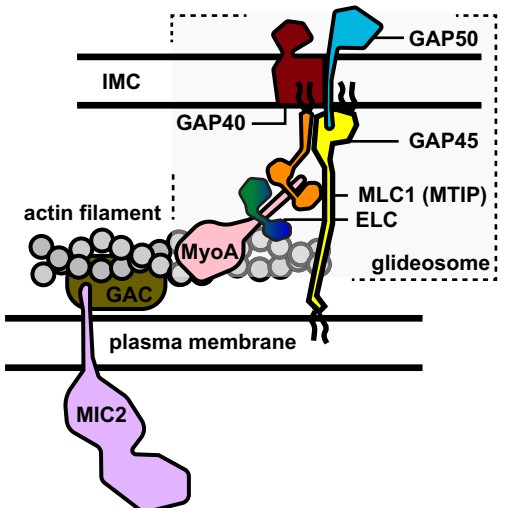

**Fig. 1 Scheme of the glideosome.** Schematic representation of the current model of the glideosome and its localization in the *T. gondii* intermembrane space. Actin polymerization occurs between the plasma membrane (PM) and the inner membrane complex (IMC) whereas myosin A is part of the glideosome, which binds the essential light chains ELC and myosin light chain MLC1 (called myosin tail interacting protein, MTIP, in *Plasmodium spp.*). Myosin A and its light chains further interact with glideosome associated proteins GAP40, GAP45 and GAP50, which anchor the glideosome in the outer membrane of the inner membrane complex. On the other side, glideosome associated connector (GAC) facilitates the association of actin filaments with surface transmembrane proteins such as MIC2.

**Table 1 Overview of thermodynamic constants measured by ITC.**

**Dimeric interactions**

| Protein (cell) | MyoA peptide (syringe) | Molar ratio | $K_d$ (nM) | $\Delta H$ (kcal/mol) | $-T\Delta S$ (kcal/mol) |
|---|---|---|---|---|---|
| MTIP | PfMyoA-C$^{ELC}$ | 0.74 ± 0.01 | 303 ± 43 | −14.4 ± 0.4 | 5.5 |
| TgELC1 | TgMyoA-C$^{ELC}$ | 1.05 ± 0.01 | 36 ± 24 | −13.0 ± 0.2 | 3.2 |
| TgELC1 (EDTA) | TgMyoA-C$^{ELC}$ | 0.81 ± 0.01 | 57 ± 18 | −13.0 ± 0.6 | 3.4 |
| TgELC2 | TgMyoA-C$^{ELC}$ | 0.85 ± 0.01 | 39 ± 12 | −15.0 ± 0.3 | 4.5 |
| TgELC2 (EDTA) | TgMyoA-C$^{ELC}$ | 0.77 ± 0.01 | 82 ± 7 | −18.0 ± 0.1 | 8.2 |
| TgELC2$^{E10A}$ | TgMyoA-C$^{ELC}$ | 0.79 ± 0.01 | 190 ± 25 | −17.0 ± 0.3 | 8.2 |
| TgELC2$^{F79A}$ | TgMyoA-C$^{ELC}$ | 0.84 ± 0.01 | 280 ± 34 | −18.0 ± 0.3 | 9.5 |
| TgELC2$^{S101A}$ | TgMyoA-C$^{ELC}$ | 0.88 ± 0.02 | 280 ± 85 | −18.0 ± 0.8 | 9.3 |
| TgELC2$^{S102A}$ | TgMyoA-C$^{ELC}$ | 0.79 ± 0.01 | 76 ± 26 | −16.0 ± 0.5 | 6.6 |
| TgELC2$^{S102E}$ | TgMyoA-C$^{ELC}$ | 0.77 ± 0.01 | 140 ± 26 | −18.0 ± 0.3 | 8.9 |
| TgELC2$^{E10A+H110A}$ | TgMyoA-C$^{ELC}$ | 0.75 ± 0.02 | 1100 ± 220 | −21.0 ± 0.9 | 12.0 |

**Trimeric interactions**

| Pre-complex with MyoA-C (cell) | Protein (syringe) | Molar ratio | $K_d$ (nM) | $\Delta H$ (kcal/mol) | $-T\Delta S$ (kcal/mol) |
|---|---|---|---|---|---|
| MTIP | PfELC | 0.86 ± 0.01 | 109 ± 6.2 | −13.4 ± 0.1 | 4 |
| MTIP | PfELC$^{S127D}$ | 0.81 ± 0.01 | 260 ± 26 | −12.6 ± 0.2 | 4 |
| TgELC1 | MLC1 | 0.92 ± 0.01 | 4.7 ± 2.5 | −39.1 ± 0.8 | 28 |
| TgELC2 | MLC1 | 0.81 ± 0.01 | 0.6 ± 0.1 | −47.6 ± 0.1 | 35 |
| TgELC2$^{R17A}$ | MLC1 | 0.92 ± 0.01 | 4.6 ± 0.4 | −49.7 ± 0.2 | 38 |
| TgELC2$^{E22A}$ | MLC1 | 0.92 ± 0.01 | 5.2 ± 1.9 | −45.9 ± 0.7 | 35 |
| TgELC2 | MLC1$^{K168A}$ | 0.79 ± 0.01 | 1.2 ± 0.8 | −47.7 ± 0.2 | 36 |
| TgELC2 | MLC1$^{Q169A}$ | 0.89 ± 0.01 | 2.3 ± 1.9 | −48.8 ± 0.5 | 37 |
| TgELC2 | MLC1$^{N172A}$ | 0.84 ± 0.01 | 4.3 ± 4.3 | −41.6 ± 0.9 | 30 |

The thermodynamic parameters were fitted by a one site binding model with the MicroCal PEAQ-ITC Analysis Software.

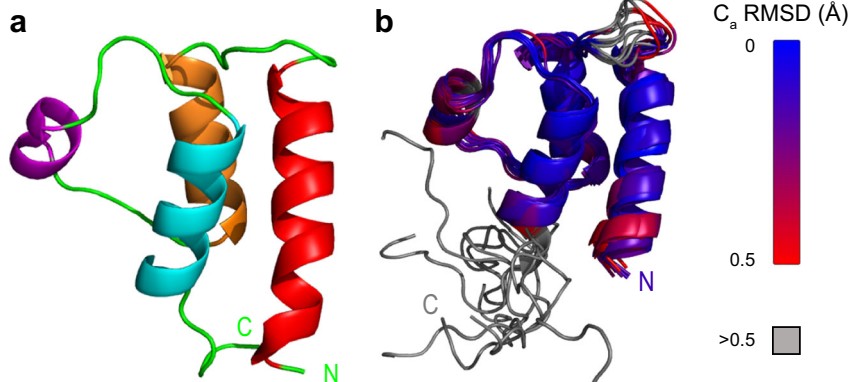

**Fig. 2 Crystal structure and NMR structures of PfELC N-terminal domain. a** Crystal structure of the N-terminal domain of PfELC, residues 1–68. PfELC displays a typical calmodulin fold with two helix-loop-helix motifs. The degenerated EF hand loops do not bind any ion. In agreement with the NMR data of full length PfELC, the protein consists of four α-helices (from N terminus red, orange, violet, cyan, loops and disordered regions in green). **b** Ten lowest-energy NMR structures of PfELC (residues 1–74, all atom RMSD of 1.23 Å) colored from lowest (blue) to highest (red) backbone RMSD compared to the crystal structure show that the loop of the PfELC first EF hand (residues 16–22) and the third helix (residues 40–47) display a certain degree of flexibility.

chemical shift analysis revealed that the protein consists of an α-helical N-terminal domain, while the C-terminal part is disordered (Supplementary Fig. 2g). Based on this finding, we were able to determine the structure of the N-terminal PfELC fragment (amino acids 1–74, PfELC-N; see Supplementary Fig. 1b) by both X-ray crystallography to 1.5 Å resolution (Fig. 2a, Table 2) and by NMR spectroscopy (Fig. 2b, Supplementary Table 4). The lowest energy NMR conformers are very similar to the crystal structure, with an average backbone RMSD of 1.4 Å over residues 1–68. The N-terminal domain of PfELC has a typical calmodulin fold with two EF-hands formed by two helix-loop-helix motifs. Both EF-hands lack the canonical residues that usually bind calcium in calmodulins[28] and in agreement with that, we did not observe any electron

density corresponding to a bound ion. PfELC-N crystallized as a dimer covalently linked via disulfide bridge, but both NMR and non-reducing SDS-PAGE indicate that the protein exists as a monomer in solution (Supplementary Fig. 1i) and the scattering data calculated from a protein monomer structure fit the measured X-ray scattering profile with $X^2 = 1.37$ (Supplementary Fig. 2h). A comparison between the crystal and the NMR structure highlights that the loop of the first EF hand (residues 16–22) and the third helix (residues 40–47) display the highest degree of flexibility, in agreement with the heteronuclear NOE experiment (Supplementary Fig. 2g, j). In general, the assigned backbone resonances in the NMR spectra superimpose for both full-length protein PfELC and the N-terminal domain, proving that the N-terminal domain maintains the

**Table 2 X-ray data collection and refinement statistics.**

|  | PfELC-N | Complex 1 | Complex 1f | Complex 2 | *P. falciparum* complex[a] |
|---|---|---|---|---|---|
| Data collection |  |  |  |  |  |
| Space group | P 21 21 21 | P 41 | P 41 | I 21 21 21 | P 43 |
| Cell dimensions |  |  |  |  |  |
|   a, b, c (Å) | 30.24, 57.51, 86.34 | 87.32, 87.32, 56.75 | 86.13, 86.13, 53.7 | 84.63, 93.48, 108.15 | 211.88 211.88 75.46 |
|   $\alpha, \beta, \gamma$ (°) | 90, 90, 90 | 90, 90, 90 | 90, 90, 90 | 90, 90, 90 | 90, 90, 90 |
| Resolution (Å) | 47.86-1.50 (1.55-1.50) | 47.58-2.39 (2.48-2.39) | 40.94-2.00 (2.07-2.00) | 40.96-2.30 (2.38-2.30) | 47.42-2.51 (2.58-2.51) |
| $R_{merge}$ | 0.03382 (0.495) | 0.106 (1.599) | 0.0431 (1.35) | 0.08044 (1.007) | 0.0874 (3.79) |
| $I / \sigma I$ | 17.68 (2.06) | 19.06 (1.40) | 31.02 (1.74) | 13.84 (1.79) | 12.69 (0.55) |
| Completeness (%) | 99.0 (98.0) | 99.9 (99.7) | 99.9 (99.4) | 99.9 (99.9) | 83.5 (8.0) |
| Total no. reflections | 104329 (9981) | 226789 (23610) | 373831 (34891) | 124597 (12474) | 768342 (75189) |
| Redundancy | 4.2 (4.2) | 13.3 (13.7) | 13.5 (12.9) | 6.4 (6.6) | 6.7 (6.6) |
| Refinement |  |  |  |  |  |
| Resolution (Å) | 1.5 | 2.4 | 2.0 | 2.3 | 2.5 |
| No. reflections | 104329 | 226789 | 373831 | 124597 | 768342 |
| $R_{work} / R_{free}$ | 0.167/0.193 | 0.189/0.231 | 0.190/0.225 | 0.186/0.219 | 0.200/0.238 |
| No. atoms | 1319 | 2523 | 2610 | 2687 | 12968 |
|   Protein | 1126 | 2457 | 2458 | 2578 | 12965 |
|   Ligands | n.a. | 2 | 5 | 33 | n.a. |
|   Solvent | 193 | 64 | 147 | 76 | 3 |
| B-factors | 36.5 | 78.2 | 65.4 | 65.3 | 99.69 |
|   Proteins | 34.8 | 78.3 | 65.4 | 65.0 | 99.69 |
|   Ligands | n.a. | 96.6 | 111 | 94.2 | n.a. |
|   Solvent | 46.5 | 72.8 | 63.2 | 62.6 | 70.51 |
| R.m.s. deviations |  |  |  |  |  |
|   Bond lengths (Å) | 0.013 | 0.008 | 0.003 | 0.007 | 0.015 |
|   Angles (°) | 1.16 | 0.97 | 0.60 | 0.87 | 2.04 |

[a]The native data of the *P. falciparum* complex were subjected to anisotropic scaling and truncation. Without truncation, $I/\sigma I$ of the native data set used for refinement falls below 2.0 between a maximum resolution of 2.75 and 2.70 Å at an overall completeness of over 99%.

same structure as in the full-length context (Supplementary Fig. 2j). These results show that isolated PfELC is monomeric in solution and adopts a calmodulin-like N-terminal fold and differs from TgELCs with a disordered C-terminal region.

**Essential light chains bind conserved sequence of MyoA.** Based on the available literature, *T. gondii* TgELCs and *P. falciparum* PfELC bind to different sites of the MyoA C-terminus[13,14,17]. For PfELC, two binding sites at the PfMyoA C-terminus (PfMyoA residues 786–803 and 801-818) were identified[14], while only one distinct binding site was experimentally confirmed for TgELCs (TgMyoA 775–795; see Fig. 3a and Supplementary Fig. 1c)[15,16]. To resolve this discrepancy, we measured the binding affinity of TgELC1, TgELC2 and PfELC to peptides that correspond to the proposed MyoA binding sites.

Both TgELC1 and TgELC2 bound TgMyoA-C^ELC (residues 777-799, see Supplementary Fig. 1c) with high affinity (36 ± 24 nM and 39 ± 12 nM, respectively), in agreement with the previously published data (Fig. 3b, c, Table 1). Strikingly, we could not monitor any binding of PfELC to the previously described binding sites but observed precipitation upon mixing PfELC with the respective peptides. Therefore, we hypothesized that the ELC binding sites are conserved between *T. gondii* and *P. falciparum* (see conserved MyoA region in Fig. 3a) and extended the PfMyoA peptide based on homology with the binding site of TgMyoA. However, precipitation occurred again and we speculated that in *P. falciparum*, the presence of MTIP bound to PfMyoA is a prerequisite for PfELC binding. Thus, we first formed a dimeric complex between MTIP and the PfMyoA neck region peptide (PfMyoA-C, residues 775–816; Fig. 3a and Supplementary Fig. 3c) and then titrated this pre-complex to

PfELC. This time, PfELC bound to the dimeric pre-complex with an affinity of 109 ± 6.2 nM (Fig. 3d and Table 1). These results indicate a particular order in which the *P. falciparum* light chains bind to PfMyoA: MTIP has to interact first and only then PfELC can bind. This is in agreement with previous reports, highlighting that PfELC co-expressed with full-length PfMyoA in insect cells can only be co-purified when MTIP is co-expressed as well[17]. On the other hand, *T. gondii* light chains showed an inverse behavior. We observed no precipitation upon binding of TgMyoA peptides to TgELCs and we were able to further titrate in MLC1 to form the trimeric complexes with high affinity (4.7 ± 2.5 nM and 0.6 ± 0.1 nM, respectively) (Fig. 3e, f and Table 1). However, the addition of MLC1 to TgMyoA peptides caused precipitation. It remains to be investigated whether the different order of binding events required for the formation of the trimeric complexes in *T. gondii* and *P. falciparum in vitro* play any role in vivo. We have demonstrated that the MyoA binding sites are conserved and topologically identical trimeric complexes form in both apicomplexan species.

**TgELCs form similar complexes with TgMyoA and MLC1.** The successful formation of trimeric assemblies of MyoA with its light chain proteins allowed us to crystallize and determine the structures of the following complexes: (i) *T. gondii* MLC1/TgMyoA-C/TgELC1 complex at 2.4 Å resolution (hereafter named complex 1) and (ii) *T. gondii* MLC1/TgMyoA-C/TgELC2 complex at 2.3 Å resolution (hereafter named complex 2) (Fig. 4a, b, d, e, Table 2). Both complexes constitute a similar architecture. TgMyoA folds into an extended α helix with a characteristic kink between residues 801–803 (angle of 139° in complex 1 and 137° in complex 2). Both TgELCs display a typical calmodulin fold with

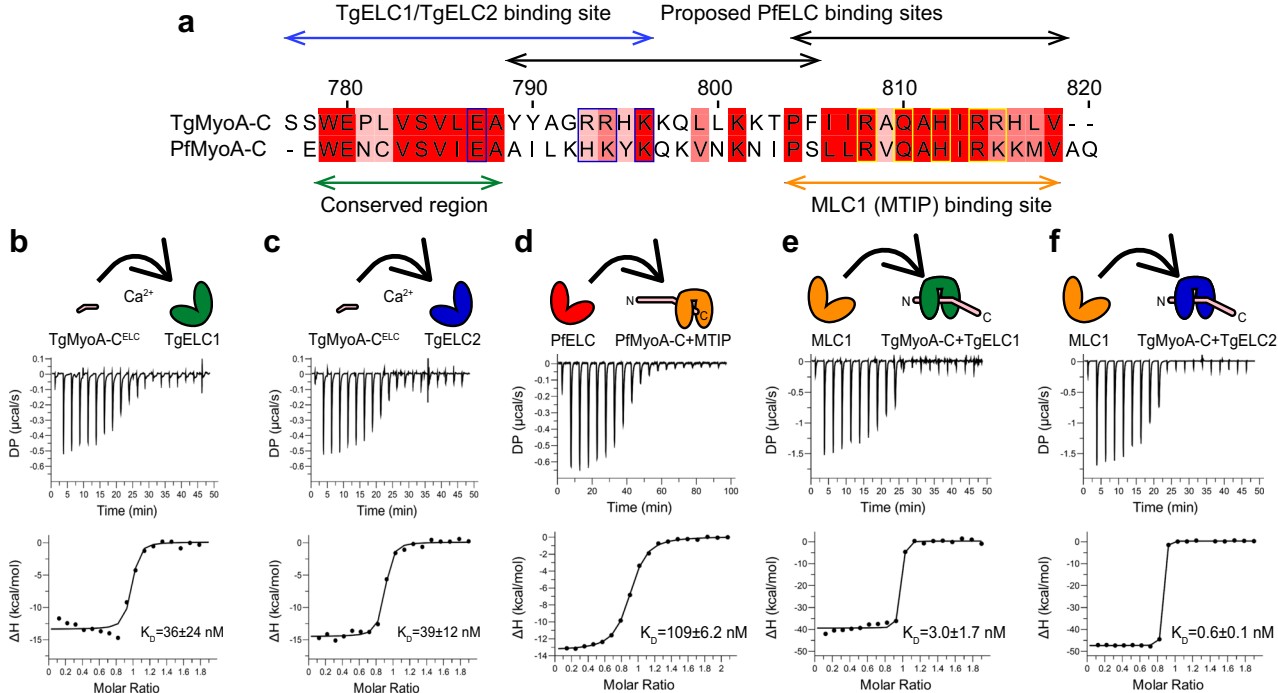

**Fig. 3 Assembly of glideosome sub-complexes in *T. gondii* and *P. falciparum*. a** Sequence comparison of TgMyoA and PfMyoA C-termini shows a conserved region (green arrow) upstream of the MLC1 (MTIP) binding site. Whereas two binding sites of PfELC at the very C-terminus of PfMyoA were proposed (black arrows)[14], our data show that the actual binding site of PfELC encompasses the MyoA conserved region and is similar to the TgELC/ TgMyoA binding site (blue arrows). The blue boxed residues indicate residues involved in polar interactions with TgELC1 and TgELC2, while yellow boxed residues form polar interactions with MLC1 (see Fig. 4 and Supplementary Data). **b, c** Isothermal titration of TgMyoA-C$^{ELC}$ with TgELC1 and TgELC2 show that both dimeric complexes form with nanomolar affinity. The upper panel shows the signal recorded directly after each injection of TgELC1 and TgELC2 and represents the thermal power that has to be applied to maintain a constant temperature in the sample cell during recurring injections. In the lower panel, the integrated heats are plotted against the peptide/protein concentration ratio. The thermodynamic binding parameters were obtained by nonlinear regression of the experimental data using a one-site binding model. **d** Binding isotherm of PfELC titrated to the preformed MTIP/PfMyoA-C complex proves that the conserved hydrophobic region of MyoA is indispensable for ELC binding. **e, f** Binding isotherms of MLC1 titrated into the pre-complex of TgMyoA-C with TgELC1 and TgELC2. MLC1 binds the pre-complex with high nanomolar affinity. All thermodynamic parameters derived from ITC measurements are summarized in Table 1.

one N-terminal and one C-terminal lobe, with each lobe comprising two EF hands. Clear additional electron density was visible only in the first EF hand of each complex and assigned to a calcium ion coordinated in a tetragonal bipyramidal geometry. Both TgELCs form conserved polar interactions with TgMyoA, involving TgMyoA residues E787, R793, R794 and K796, a π-π stacking interaction between the conserved residue pair W779-F79 and a group of hydrophobic residues clustered around the conserved TgMyoA region P801-Y810 (Fig. 4b, e, Supplementary Data). Mutational analysis on TgELC2 (Table 1, Supplementary Fig. 3a) showed that disrupting one of the polar interactions or the conserved π-π stacking interaction W779-F79 has only a moderate effect on the binding affinity of TgMyoA to TgELC2 and suggests that the hydrophobic residues in the conserved MyoA region play a crucial role for complex formation. In agreement, the phosphomimetic mutation of residue S102, previously shown to be phosphorylated[29], only had a moderate effect on the affinity of TgELC2 to the MyoA peptide, indicating that a single phosphorylation event is likely not sufficient to regulate complex formation (Table 1 and Supplementary Fig. 3b). Complexes 1 and 2 are monomeric in solution, but while the calculated scattering data of complex 1 fit the experimental scattering data with a $X^2 = 1.26$, the structure of complex 2 displays a higher $X^2 = 2.41$, indicative of small structural differences in solution (Fig. 4c, f, Supplementary Table 3). Taken together, *T. gondii* TgELCs form tight complexes with MyoA and MLC1 and the corresponding binding interfaces are dominated by hydrophilic and hydrophobic interactions.

**PfELC binds PfMyoA in a structurally distinct manner**. To investigate whether the homologous complexes from *T. gondii* and *P. falciparum* are structurally similar, we determined the crystal structure of the *P. falciparum* trimeric complex (PfMyoA, MTIP, PfELC) at 2.5 Å resolution (Fig. 4g, Table 2). Overall, this structure resembles a similar fold and conformation compared to the *T. gondii* trimeric complexes, with the typical MyoA helix kink of 131° between the MTIP and PfELC binding sites. While the secondary structure elements are maintained, the position of the PfELC helices differ. The N-terminal lobe of PfELC aligns well to TgELCs structures (backbone RMSD of 2.5 Å to TgELC1), but the C-terminal lobe adopts a different orientation with respect to the MyoA helix (backbone RMSD of 3.5 Å to TgELC1), resulting in a reduced number of polar interactions between PfELC and PfMyoA (Supplementary Fig. 3e). This explains the lower binding affinity of trimeric complex formation in *P. falciparum* (Fig. 4h, Table 1). Of note, the electron density of the PfELC C-terminal lobe is less well defined compared to the remaining structure, which is likely caused by increased flexibility of the C-terminal loop of PfELC. This is also reflected in the comparison of the calculated scattering data from the *P. falciparum* complex structure and the recorded SAXS data with $X^2 = 3.9$ (Fig. 4i, Supplementary Table 3).

Due to the reduced number of interacting residues at the PfELC C-terminus (Supplementary Data), it seems plausible that C-terminal phosphorylation could play a regulatory role in binding of PfELC to PfMyoA. To test this hypothesis in vitro, we

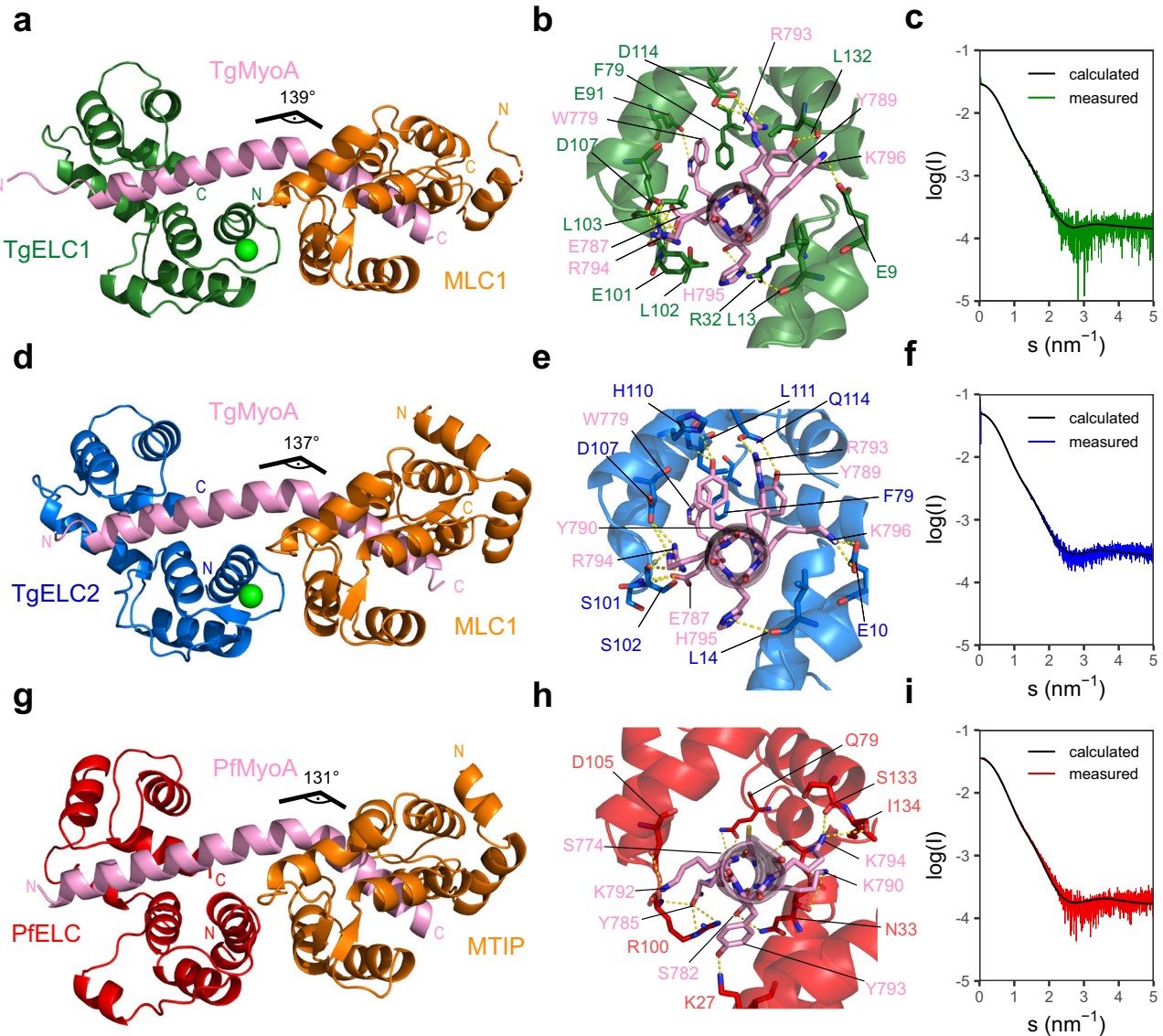

**Fig. 4 X-ray structures of trimeric glideosome sub-complexes. a, d, g** Crystal structures of trimeric complex of TgELC1 (green) or TgELC2 (blue) or PfELC (red) with MyoA-C (pink) and MLC1/MTIP (orange). The complexes are topologically similar and the MyoA helix displays a characteristic kink between residues 801–803. ELCs bind upstream of the MLC1/MTIP binding site. **b, e, h** Binding interface between MyoA-C (pink) and TgELC1 (green) in complex 1, TgELC2 (blue) in complex 2 or PfELC (red) in *P. falciparum* complex. Residues involved in polar interactions are labeled with the corresponding color and shown in stick representation. Most polar interactions are mediated by the C-terminal lobes of ELCs and the hydrophobic interactions between ELCs and the conserved hydrophobic MyoA residues play a crucial role in complex formation as evident from ITC measurements. **c, f, i** SAXS analysis of the trimeric complexes. Calculated scattering curves of complex 1, complex 2 and *P. falciparum* complex fit the respective experimental data with $\chi^2$ equal to 1.26, 2.41 and 3.9, respectively.

mutated residue S127, that has previously been shown to be phosphorylated in vivo[29], to a phosphomimetic aspartate residue and observed that the affinity for this variant to PfMyoA C-terminal peptide dropped twofold (Table 1 and Supplementary Fig. 3d). S127 does not directly interact with PfMyoA, but forms a polar interaction with PfELC residue N75, maintaining the tertiary structure of the C-terminal lobe. Based on available data, it is likely that phosphorylation of S127 has a direct impact on the interaction of PfELC with PfMyoA, however, in vivo experiments are necessary to study the impact of this phosphorylation on the glideosome assembly and function.

**ELCs induce α-helical structure in MyoA.** Previous reports have shown that the presence of *P. falciparum* and *T. gondii* essential light chains increase the speed of the myosin A motor

twofold[14,16,17]. To understand the function of ELCs on a molecular level, we characterized TgELC2 in a free and bound state with TgMyoA-C^ELC (see Supplementary Fig. 1c). On size exclusion chromatography, the dimeric complex of TgELC2 and TgMyoA-C^ELC elutes later than TgELC2 alone, indicating that the hydrodynamic radius of TgELC2 decreases upon binding of TgMyoA-C^ELC (Fig. 5a). To quantify the structural changes upon binding, we compared the parameters calculated from the SAXS data of TgELC2 alone and in complex with TgMyoA-C^ELC (Fig. 5b, c, Supplementary Fig. 4a, b, Supplementary Table 3). Changes in the dimensionless Kratky plot (Fig. 5b) as well as the drop of the radius of gyration (2.15 nm to 1.73 nm) and maximum particle size (6.7 nm to 5.5 nm, Fig. 5c) highlight that the dynamic TgELC2 protein undergoes compression upon interaction with the TgMyoA C-terminus. This rigid conformation

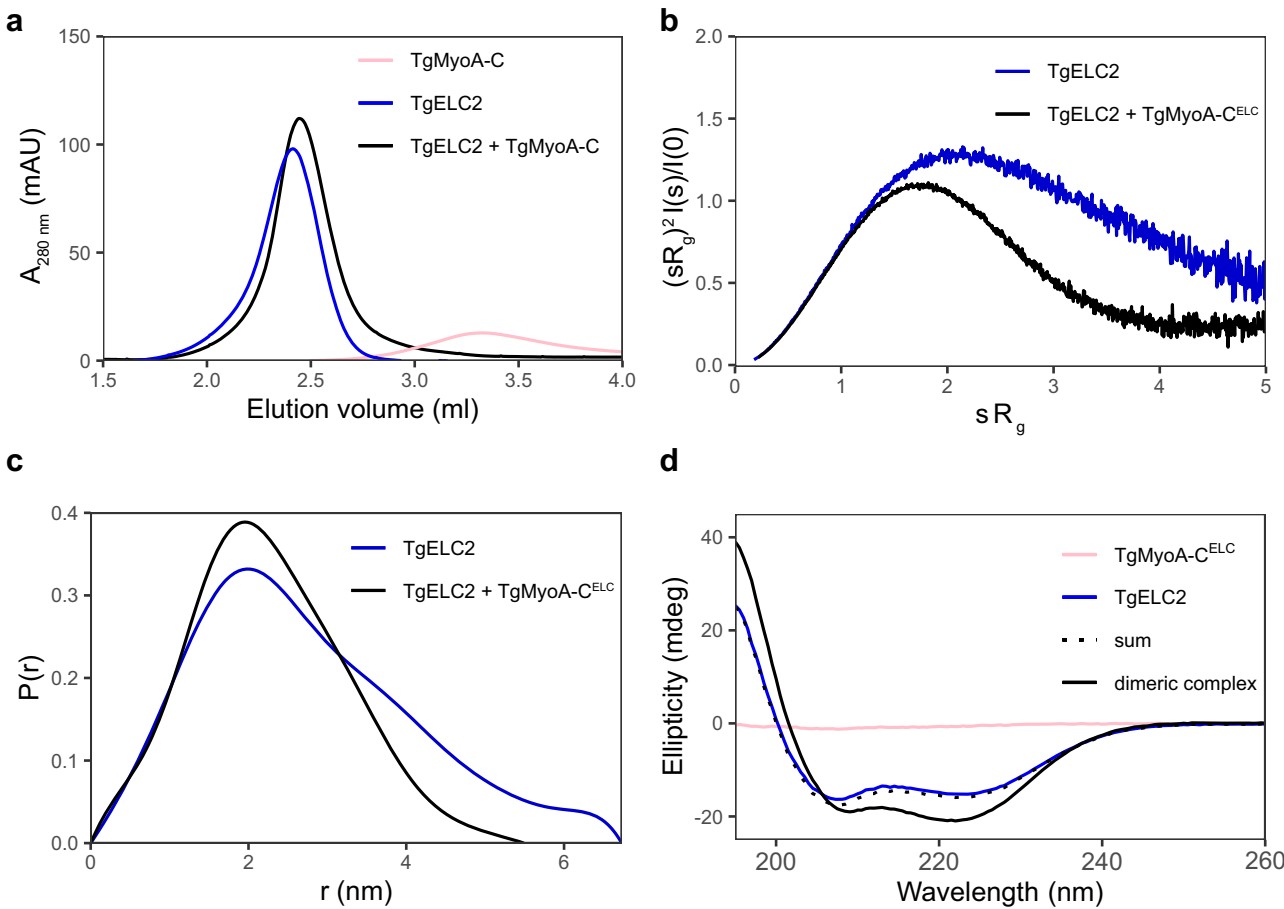

**Fig. 5 TgELCs and TgMyoA undergo large conformational changes upon binding. a** Dimeric complex of TgELC2 and TgMyoA-C elutes at longer retention times than isolated TgELC2 on Superdex 200 5/150 column, suggesting that the hydrodynamic radius of TgELC2 decreases upon TgMyoA-C binding. **b** Dimensionless Kratky plots of isolated TgELC2 and in complex with TgMyoA-C. The plot of TgELC2 in complex with TgMyoA-C$^{ELC}$ (black) has a maximum close to $sR_g = \sqrt{3}$ and converges to zero, unlike isolated TgELC2 (blue), suggesting that TgELC2 in isolation is rather extended and compacts upon binding to TgMyoA. **c** The distance distribution calculated by Guinier analysis from the SAXS data further confirms that TgELC2 undergoes compaction upon TgMyoA binding. TgELC2 displays wider distance distribution with $d_{max} = 6.7$ nm, whereas the distance distribution of the dimeric complex is narrower with $d_{max} = 5.5$ nm. **d** The far-UV CD data indicate that TgELC2 induces a α-helical structure in TgMyoA upon binding. The individual spectra of TgELC2 and TgMyoA-C$^{ELC}$ do not sum up to the CD spectrum of their dimeric complex and the CD spectrum of the dimeric complex displays more pronounced features of α-helical secondary structure with lower ellipticity at 222 nm and higher ellipticity at 195 nm compared to the sum of individual components. CD spectra were recorded in a 1 mm cuvette at a concentration of 5 μM of each component in 10 mM NaP (pH 7.5), 150 mM NaF and 0.25 mM TCEP at 20 °C.

allows the neck region to act as the lever arm of myosin and its stiffness directly correlates with the myosin step size and speed[30–32]. Although our crystal structures show that TgMyoA-C forms a continuous α helix, we noticed that both TgMyoA-C as well as PfMyoA-C are unfolded or partially unfolded in the absence of binding partners (Supplementary Fig. 4c). Indeed, the C-terminal amino acid residues of the recently published TgMyoA[26] and PfMyoA[27] motor domain structures could not be resolved, likely due to their intrinsically disordered nature. We hypothesized that the essential light chains can induce the formation of an α-helical structure in MyoA upon binding. Therefore, we measured far-UV CD spectra of TgMyoA-C$^{ELC}$ and TgELC2 in isolation and in complex (Fig. 5d). The data revealed that TgMyoA-C$^{ELC}$ is predominantly unstructured while TgELC2 has an α-helical fold. However, the CD spectrum of the dimeric complex displays a markedly higher α-helical content than the sum of the spectra of the two individual components, suggesting that the content of the α-helical secondary structure increased upon formation of the complex. We also observed a similar, albeit less pronounced effect for the TgELC1-TgMyoA-C$^{ELC}$ and *P.*

*falciparum* trimeric complex assembly (Supplementary Fig. 4d, e). We anticipate that the increase in α-helical secondary structure content corresponds to the induction of the structure of the TgMyoA C-terminus, which in turn stiffens the TgMyoA lever arm. As a result, the myosins are capable of undergoing a larger step size and thus increase their speed, in agreement with the published functional measurements for both *T. gondii* and *P. falciparum* myosin A motors[14,16,17].

**Calcium stabilizes but has no impact on complex assembly.** The myosin light chains together with the myosin heavy chain neck region constitute a regulatory domain that influences the biochemical and mechanical properties of myosins either upon phosphorylation[33–36] or by direct binding of calcium[37,38]. Apicomplexan invasion is a tightly regulated process, which involves an increase in intracellular calcium concentration[39]. To investigate the role of calcium bound in the first EF hand of both TgELCs, we determined an additional crystal structure of the calcium-free complex TgELC1/MLC1/MyoA-C at 2.0 Å resolution (complex 1f, Fig. 6a, Table 2). Complex 1f generally adopts

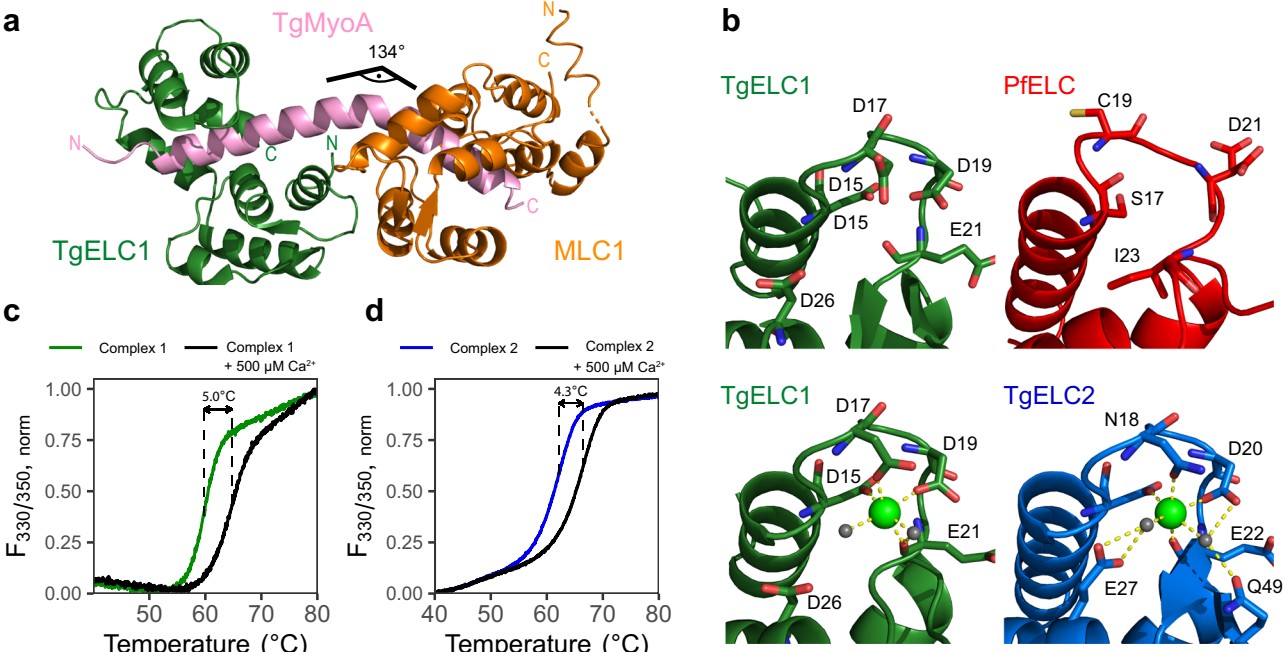

**Fig. 6 Role of calcium in TgELCs. a** Crystal structure of the glideosome trimeric complex composed of TgELC1 (green), MLC1 (orange) and TgMyoA-C (pink) in the absence of calcium (complex 1f). The absence of calcium does not cause a major structural rearrangement (see Fig. 5a). **b** Structural comparison of the first EF hand in ELCs and calcium coordination between complex 1f, complex 1, complex 2 and PfELC-N. Whereas PfELC does not bind any ion due to a degenerated sequence in its EF hand, both TgELCs in complex 1 and complex 2 bind calcium in a tetragonal bipyramid coordination, including two water molecules. These water molecules are further stabilized in complex 2 by additional residues (E27, Q49, D20). In complex 1f, the side chain of residue D17 is flipped by 120°, enabling the release of calcium. **c, d** Thermal stability change of trimeric complex 1 and complex 2 upon addition of calcium measured by nanoDSF. The stability of both complexes strongly increases upon calcium binding.

the same conformation as complex 1. The MyoA-C helix is kinked at a similar angle (134°), and the binding interfaces between MLC1 and TgMyoA as well as between TgELC1 and MyoA are identical to complex 1 (Supplementary Data). The first EF hand loop and the calcium binding residues remain in the same conformation as in complex 1 except for the side chain of aspartate 17 which is flipped by 120° and thereby enables the release of calcium from the binding pocket (Fig. 6b). In complex 1, calcium is coordinated in a tetragonal bipyramidal geometry by the carboxyl groups of side chains D15, D17, D19, the carbonyl group of E21 and two water molecules. In complex 2, calcium is similarly coordinated by the homologous side chain residues of D16, N18, D20, the carbonyl group of E22 and two water molecules. Additionally, in complex 2, these water molecules are further stabilized by interactions with the side chains of E27 and Q49. Contrary to *T. gondii* TgELCs, the homologous EF hand loop of PfELC (in isolation or in complex) is bent to the other side and does not possess the residues needed for coordination of calcium (Fig. 6b). In agreement with the presented crystal structures, calcium has no major influence on the secondary structure of individual TgELCs or PfELC (Supplementary Fig. 5a).

Powell et al. recently showed that the absence of calcium notably reduces the affinity of TgELC1 for the MyoA C-terminus[15]. To investigate this effect in both *T. gondii* essential light chains, we measured the affinity of TgELC1 and TgELC2 to the TgMyoA peptide with wild type proteins either in the presence of 5 mM calcium or 5 mM EDTA. Strikingly, the difference in affinity is only minor in both cases, with an observed twofold decrease in affinity in the presence of 5 mM EDTA compared to 5 mM calcium (Supplementary Fig. 5b). This is rather surprising, considering the fact that the regulatory role of calcium has been proposed for other myosin light chains[37,40].

Our binding data are supported by the available crystal structures, where a clear role for calcium regulation is not directly evident. While the presence of calcium affects the affinity of ELCs only to a minor extend, we observed a pronounced effect of calcium ions on the thermal stability of the trimeric complex in a concentration dependent manner (Fig. 6c, d, Supplementary Fig. 5c). This reveals that calcium ions bind TgELCs and mediate substantial stabilization of their sub-complexes, although they do not markedly change their structure or affinity. This is in agreement with previously published functional data, reporting that the absence of calcium does not alter the function of the myosin A motor in both *P. falciparum*[17] and *T. gondii*[13]. It is likely that the presence of calcium could have a rather indirect effect, for example by modulating the activity of kinases which in return change the phosphorylation status of members of the glideosome[41–43]. In conclusion, calcium binding by the first EF hand of TgELCs does not structurally impact the formation of the complex but increases the stability of the complexes per se.

**Light chain interactions do not trigger structural changes**. Based on our structural work, we have shown that the formation of the TgMyoA-TgELCs dimeric complexes leads to large structural changes and folding of the MyoA C-terminus. In the trimeric complexes, interactions between the light chains have been proposed to mediate the transmission of regulatory signals from distal (MLC) to proximal light chain (ELC) light chains[33]. To assess the structural changes that could result from the interaction between the two light chains, we recorded SAXS data of the TgELC2-TgMyoA-C$^{ELC}$ dimeric complex and compared them to the scattering profile calculated from complex 2 without MLC1 (Supplementary Fig. 6a). Based on a resulting $X^2$ of 1.16 Å, it is unlikely that TgELC2 undergoes structural changes upon trimeric complex formation. Similarly, MLC1 and MTIP adopt the same

**Fig. 7 Light chain interactions upon trimeric complex formation.** Binding interfaces between TgELCs and MLC1 in the trimeric complex structures of (from left) complex 1f, complex 1 and complex 2. Corresponding residues are labeled with the respective color. The same set of residues (K168, Q169, N172, Y177) is involved in polar interactions (indicated by yellow dashes) on the MLC1 site, but various residues are utilized by TgELCs.

conformation as in already described structures of their dimeric complexes with MyoA (PDB IDs 5vt9 and 4aom, respectively) and the key interactions remain unperturbed in the presence of ELCs (Supplementary Fig. 6e–g, Supplementary Data).

Thus, light chains do not exhibit any major structural rearrangements upon trimeric complex formation, although the crystal structures revealed a small interaction surface between both light chains formed by several polar interactions near the ELC calcium binding site. These interactions were previously proposed to be only present when calcium is bound[16]. However, the calcium-free crystal structure shows that these interactions are rather independent of the presence of calcium and conserved between complex 1 and complex 2 (Fig. 7). We additionally performed mutational analysis of the interacting residues at the interface of MLC1 and TgELC2. We observed only a minor decrease in affinity upon mutation, but the measured affinities reached the limitations of reliable high affinity ITC measurements (Table 1 and Supplementary Fig. 6d). This leaves open the possibility of cross-talk between the two light chains, however, we do not expect these to have a large impact on the overall structure and myosin motor function because the effect of the mutations at the light chain interface is only minor.

To complete our analysis, we examined whether the formation of the trimeric complexes impacts the structure of the MLC1 N-terminus. The disordered N-termini of MLC1 and MTIP are of particular interest because they are expected to anchor myosin A to the IMC via interaction with GAP45[10]. Our SAXS data reveal that the trimeric complex containing full-length MLC1 displays a notably larger maximum particle size ($D_{max}$ = 14 nm) and radius of gyration (3.50 ± 0.02 nm) in comparison to the complex used for crystallization (Fig. 4d, Supplementary Fig. 6b and Supplementary Table 3), indicating that the MLC1 N-terminus remains disordered even in the trimeric complex with TgMyoA and TgELC1.

Next, we explored the stretch of MLC1 residues 66–77, which are on the border of the disordered N-terminus and the structured domains. The electron density in complex 2 reveals an additional α helix for this area, which is absent in complex 1, suggesting that residues 66–77 are disordered. We hypothesized that in solution, this helix is in equilibrium with a disordered state. To investigate this possibility, we compared the distance distributions calculated from SAXS data measured on complex 1 using MLC1 constructs spanning residues 66–210 or 77–210. In case residues 66–76 form exclusively an α helix in solution, we expect them to fold back towards the center of the molecule and the maximum particle distance $D_{max}$ should stay identical. However, $D_{max}$ in the trimeric complex with MLC1$^{77–210}$ (8.2 nm) is markedly lower compared to the construct containing residues 66–76 (9.5 nm with MLC1$^{66–210}$, see Supplementary

Fig. 6b). Moreover, SAXS data of the trimeric complex with MLC1$^{66–210}$ agree less with the corresponding crystal structure than with the shorter MLC1$^{77–210}$ construct ($X^2$ equals 1.26 vs 1.04, Supplementary Fig. 6c). The flexibility within residues 66–77 is additionally apparent from the normal mode analysis (Supplementary Fig. 7a, see below). We assume that residues 66–77 of MLC1 exist in equilibrium between α-helical and disordered conformation in solution and believe that this feature may have further implications on the function of the protein, namely anchoring MyoA to the membranes of the IMC or interacting with other members of glideosome, such as GAP45. Knowing that the stiffening of the MyoA lever arm by ELCs increases the motor activity[14,16,17], we find it unlikely that the MLC1/MTIP N-termini are disordered when assembled within the glideosome. We propose that, similarly to the MLC1 helix 67–77, the secondary structure can be induced in the entire MLC1/MTIP N-terminus upon binding to presumably GAP45, as described here for the ELC-MyoA interaction.

**TgMyoA complexes follow the dynamics of traditional myosins.** Previously reported structures of myosins in complex with their light chains suggest that the converter domains interact with the essential light chain to further stabilize the rigid lever arm and possibly transmit the structural changes from the myosin motor domain to the lever arm[44,45]. Similarly, it has been proposed that TgELC1 might constitute a small binding interface with the TgMyoA converter domain[15]. To investigate whether the crystal structures of *T. gondii* complexes are compatible with these observations and to ensure that they do not clash with the TgMyoA core, we built structural models of the TgMyoA motor and neck domain bound to MLC1 and TgELC1 or TgELC2 (Fig. 8).

In both cases, the energy-minimized models did not contain any clashes, indicating that our structures are compatible within the full-length context of TgMyoA (Fig. 8a, b). TgMyoA residues 762–818, which constitute the lever arm, maintained a continuous α helix after energy minimization, with both TgELC1 and TgELC2 forming a small number of contacts with the TgMyoA converter domain. These contacts mainly involve the side chain of arginine 81 of TgELC1 or TgELC2 and residues 720–724 of TgMyoA, which is in agreement with the previously published HDX data[15]. To further explore the dynamics of full-length TgMyoA with its light chains, we performed normal mode analysis in an all-atom representation on five energy-minimized models from complex 1 and complex 2, and subsequent deformation analysis which allowed us to identify potential hinge regions within these structures. In both cases, all five reconstructed models displayed nearly identical pattern of motions (see Supplementary Fig. 7a for complex 2): the structures undergo

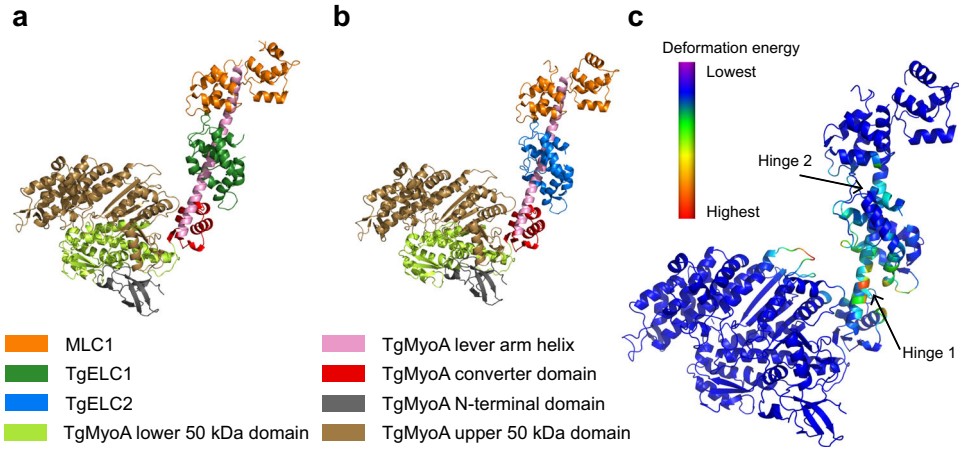

**Fig. 8 Trimeric complexes modeled in the full-length MyoA context. a** Energy-minimized model of complex 1 as a part of TgMyoA. **b** Energy-minimized model of complex 2 as a part of TgMyoA. The models show that the crystal structures of the trimeric complexes are compatible with the structure of TgMyoA and maintain the α-helical structure of the TgMyoA lever arm. No clashes between TgMyoA and TgELCs were observed. **c** Deformation analysis of complex 2 identified two hinge regions in the lever arm of myosin A, which contribute to most of the observed dynamics of the protein complex within the 20 lowest-energy modes. The model is colored by deformation energy from lowest (violet) to highest (red). The hinges localize to the TgMyoA lever arm between the converter domain and the TgELC2 binding site (hinge 1, residues 773–777) as well as between the TgELC2 and MLC1 binding sites (hinge 2, residues 799–801). These deformations agree with the role of TgMyoA in the pre-power stroke state in the context of a power stroke cycle, where the myosin is probing the conformational space to bind to actin.

bending in the hinge region of TgMyoA residues 773–777 in two perpendicular directions (mode 7 and 8) as well as twisting in the same region (mode 9). In the remaining modes (modes 10 and higher), the movement further propagates throughout the lever arm helix up to TgMyoA residue 799. As a result, the deformation analysis of the 20 lowest energy modes predicts the hinge region of the TgMyoA lever arm between TgELCs and the converter domain, and an additional hinge between TgELCs and MLC1 (complex 2 in Fig. 8c and complex 1 in Supplementary Fig. 7b). Such dynamics of myosin light chains is similar to what has been previously described in conventional myosins[45,46] and the flexibility in the first TgMyoA hinge could contribute to the efficient rebinding of the myosin motor domain to actin in the pre-power stroke state (Supplementary Fig. 7c)[40]. In conclusion, the structures of the trimeric complexes composed of the TgMyoA light chains and TgMyoA C-terminus are compatible with full-length TgMyoA and exhibit dynamics that are similar to the dynamics of conventional myosins.

Finally, ELCs generally interact with the myosin converter domain and likely stabilize the hinge region of the myosin neck between the ELC and the converter domain (TgMyoA residues 775–777)[40,46]. A small interaction interface between the converter domain and TgELC1 has also been suggested previously[15]. Our models now highlight that both TgELC1 and TgELC2 form polar interactions with the converter domain, however, these are not sufficient to maintain the rigid structure, and the TgMyoA hinge between ELC and the converter domain contributes to most of the movement of the myosin complex. Nevertheless, the normal mode analysis was performed in the absence of a bound nucleotide or actin and the interface between TgELCs and the converter domain might become more rigid once TgMyoA binds actin, as has been previously described for other myosins[47].

## Discussion
Although both gliding and invasion of apicomplexan parasites have been intensively studied in the past, the lack of structural data inhibits the broader understanding of these processes on a molecular level. Our work represents a further step towards grasping glideosome function and the mechanism of apicomplexan gliding and invasion. We have determined crystal structures of the glideosome trimeric sub-complexes of two main apicomplexan representatives, *P. falciparum* and *T. gondii*. Our structures together with binding data show that ELCs bind a conserved sequence of MyoAs. The C-terminus of PfELC is disordered in isolation compared to TgELCs and also adopts a distinct position when bound to PfMyoA, compared to *T. gondii* complexes. The structures also reveal potential regulatory phosphorylation sites on ELCs and our mutational analysis indicates that phosphorylation events can decrease the ELC binding affinity. We have further investigated the role of ELCs in glideosome assembly as well as the impact of calcium ions that we have observed to be bound in the first EF hands of TgELCs. An additional calcium-free structure of a *T. gondii* trimeric sub complex shows that no major structural changes occur upon calcium binding. Indeed, we observe that calcium ions have no impact on the assembly of the complexes but rather stabilizes the trimeric complexes per se. Finally, our biophysical analysis demonstrates that ELCs undergo compression upon binding to MyoA, which induces an α helical structure and thereby stiffens the MyoA lever arms. Our functional observations explain previously published data showing that ELCs can double the speed of a myosin A motor whereas calcium has no effect. In conclusion, our study complements and rationalizes the role of glideosome components that have been previously observed while providing new structural and functional data that will be important in the future elucidation of glideosome structure and mechanism of apicomplexan gliding.

## Methods
**Cloning.** Open reading frames encoding TgELC2 (*TGME49_305050*) and TgMLC1 (*TGME49_257680*) sub cloned *via* NdeI/XhoI restriction enzymes into pET28a (+)-TEV vector were purchased from *GenScript*. The TgELC1 gene was cloned, by extending the *TGME49_269442* open reading frame (*GenScript*) into a pNIC28_Bsa4[48] vector *via* BsaI restriction sites. DNA sequences of PfELC (*PF3D7_1017500*), PfELC-N (residues 1–74), PfMTIP (*PF3D7_1246400*), PfMTIP-S (residues 60–204) and PfMTIP[77–204] were amplified from *P. falciparum* 3D7 cDNA and cloned into a pNIC28_Bsa4 vector *via* BsaI restriction sites. These constructs have an N-terminal TEV-cleavable His₆-tag. TgMLC1-S (residues 66–146) was sub cloned into a pNIC_CTHF[48] vector *via* the BfuI restriction site. The vector has a C-terminal TEV-cleavable His₆-tag and FLAG-tag. The sequence encoding TgMyoA-C was amplified by two complementary primers and cloned *via*

NcoI/KpnI restriction enzymes into a pET_GB1 vector. This construct contains an N-terminal TEV-cleavable His-GB1 domain. Expression cassettes of His-TgELC1 and His-GB1-TgMyoA-C were then sub cloned via NdeI/XbaI restriction enzymes into a pPYC[49] vector. The His-GB1-TgMyoA-C gene was then cut by SpeI/XbaI restriction enzymes and inserted into SpeI-cut pPYC-His_TgELC1 to construct the co-expression vector pPYC with TgELC1 and TgMyoA-C.

**Mutagenesis.** Site directed mutants were generated by blunt-end PCR. Briefly, the plasmids were amplified by primers which contain the alternative bases on their 5′ ends and anneal upstream and downstream of the target triplet. The PCR products were digested by DpnI (NEB) overnight at 37 °C and purified by a PCR purification kit (Qiagen). Subsequently, the 5′ ends of the PCR products were phosphorylated by T4 polynucleotide kinase (NEB), the products were purified and the free ends of the plasmid re-ligated by T4 DNA ligase (NEB). The positive clones were subsequently selected and their sequence was verified by sequencing.

**Protein expression and purification.** The proteins were overexpressed in *E. coli* BL21(DE3) (MLC1, MTIP, MTIP-S, co-expressed TgELC1-TgMyoA-C + MLC1-S) or *E. coli* BL21-CodonPlus(DE3)-RIL (TgELC1, TgELC2, PfELC, PfELC-N, MLC1-S), in TB medium. The bacterial cultures were induced at $OD_{600nm}$ of 0.6 with 1 mM IPTG and harvested after 4 h at 37 °C (TgELC1, TgELC2 or PfMTIP) or induced at $OD_{600nm}$ of 0.6 by 0.2 mM IPTG and harvested after 16 h at 18 °C (PfELC, PfELC-N, MLC1). The expression of PfELC and PfELC-N for NMR measurements was performed in minimal expression medium as described elsewhere[50].

The cell pellets were resuspended in lysis buffer (20 mM NaP (pH 7.5), 300 mM NaCl, 5% glycerol, 15 mM imidazole, 5 units/ml DNase I, 1 tablet of protease inhibitors (Roche) per 100 mL buffer, 1 mg/mL lysozyme, 0.5 mM TCEP) and the bacteria were lysed by three passages through an emulsifier (EmulsiFlex-C3, Avestin) with a maximum pressure of 10,000 psi. The lysate was centrifuged (20 min, 19,000 g) and incubated with 2 ml of Ni-IMAC beads (ThermoFisher) per 1 L of culture on a rotatory wheel (1 h, 4 RPM). The lysate was then transferred into a gravity column and washed twice with 10 mL wash buffer (20 mM NaP (pH 7.5), 300 mM NaCl, 5% glycerol, 15 mM imidazole, 0.5 mM TCEP). The bound protein was eluted with 10 mL and subsequently with 5 mL of elution buffer (20 mM NaP (pH 7.5), 150 mM NaCl, 5% glycerol, 250 mM imidazole, 0.5 mM TCEP). The elution fractions were pooled and 0.5 mg of TEV protease per liter of bacterial culture was added. The samples were dialyzed (2 kDa cut-off) against 500 mL wash buffer or, in case of PfELC and PfELC-N, against 50 mM Tris (pH 8.0), 20 mM NaCl, 0.5 mM TCEP overnight. Next day, the samples were incubated on a gravity column with 1 ml Ni-beads per 1 L of culture. The flow-through was concentrated (10 kDa cut-off) to maximum of 10 mg/mL and further purified by size exclusion chromatography on a Superdex 200 HiLoad column (GE Healthcare; PfELC, MTIP, MTIP-S, MLC1, MLC1-S) or on a Superdex 75 HiLoad column (GE Healthcare; TgELC1, TgELC2, PfELC-N, co-expressed TgELC1-TgMyoA-C), using gel filtration buffer (20 mM HEPES (pH 7.5), 150 mM NaCl, 0.5 mM TCEP). Finally, the samples were concentrated (10 kDa cut-off) up to 15 mg/mL and either directly used or flash-frozen for later use. Due to instability, PfELC was always directly used within 3 days of the purification without freezing. All steps were performed at 4 °C.

**SDS-PAGE analysis.** The concentrated samples of PfELC were dialyzed against 50 mM Tris (pH 8.0), 20 mM NaCl, and 0, 0.25, 0.5 or 1 mM TCEP overnight at 4 °C. Subsequently, the protein concentration was adjusted to 1 mg/mL and 50 μL of each sample was mixed with a fivefold excess of 2-iodoacetamide. The samples were incubated for 1 h at 37 °C and afterwards, 10 μL of each sample was mixed with 5 μL of non-reducing loading dye. The gel was run at 180 V for 40 min and stained by Direct Blue.

**Analytical gel filtration.** The proteins and protein complexes were analyzed by analytical gel filtration using a Superdex 200 5/150 column (GE Healthcare) and the 1260 Infinity Bio-inert high-performance liquid chromatography system (Agilent Technologies) at 10 °C. The system and column were equilibrated in 20 mM HEPES (pH 7.5), 150 mM NaCl, 0.5 mM TCEP and 30 μL of each sample was injected by an auto sampler. The system was run at 0.2 mL/min for 20 min and the elution profile was recorded by a UV detector.

**Thermal shift assay.** The stability of the different proteins was measured by nanoDSF (Prometheus NT.48, NanoTemper Technologies, GmbH). The proteins were first dialyzed against 1 L of gel filtration buffer supplemented with 5 mM EDTA overnight at 4 °C and subsequently 2× against 1 L of gel filtration buffer without EDTA overnight at 4 °C. The protein concentration was then adjusted to 100 μM (individually or 100 μM each component of a complex) in gel filtration buffer and varying concentrations of calcium chloride (0–500 μM). 10 μL of sample was loaded in the glass capillaries and heated from 20 °C to 95 °C with a heating rate of 1 °C/min. The fluorescence signals with an excitation wavelength of 280 nm and emission wavelengths of 330 and 350 nm were recorded and the melting temperature was calculated as either the maximum of the derivative of the ratio of fluorescence at 330 and 350 nm, or as maximum of the derivative of the fluorescence recorded at 330 nm.

**Circular dichroism.** To estimate the secondary structure content of the proteins and peptides, we measured circular dichroism on a Chirascan CD spectrometer (Applied Photophysics). For spectrum measurements, the protein or peptide concentration was adjusted to 100 μM and diluted tenfold by 10 mM NaP (pH 7.5), 20 mM NaCl, 0.25 mM TCEP just prior to the measurement. To measure the difference in secondary structure content in presence or absence of calcium, the proteins were first dialyzed against 1 L of gel filtration buffer supplemented with 5 mM EDTA overnight at 4 °C and subsequently 2× against 1 L of gel filtration buffer supplemented with ±1 mM $CaCl_2$ overnight at 4 °C. The proteins were then diluted to 5 μM or 10 μM with 10 mM NaP (pH 7.5), 20 mM NaCl, 0.25 mM TCEP and ±1 mM $CaCl_2$ just prior to the measurement. The CD spectrum was measured between 200 nm and 260 nm with 1 nm steps in triplicates using a 2 mm quartz cuvette. To assess the induction of structure in the dimeric protein complexes, each component was diluted by 10 mM NaP (pH 7.5), 150 mM NaF and 0.25 mM TCEP to a final concentration of 5 μM. The circular dichroism was measured 10× between 195 nm and 260 nm with 0.5 nm step in 1 mm quartz cuvette. The data were averaged, background subtracted and analyzed by K2D algorithm[51] using DichroWeb[52].

**Isothermal titration calorimetry.** To measure the interaction of TgELC1 or TgELC2 with the TgMyoA-$C^{ELC}$ peptide (S777-Q798), the peptides were dissolved and the proteins were dialyzed in gel filtration buffer supplemented with either 5 mM $CaCl_2$ or EDTA overnight at 4 °C and 2 μL of a 200 μM peptide solution were injected 19 times into 20 μM protein. To measure the interaction of the trimeric complex, first, the peptides were dissolved and the proteins dialyzed against gel filtration buffer supplemented with 1 mM $CaCl_2$. The complex of TgELC1, TgELC2 or MTIP-S with the MyoA peptide (S777-V818 in *T. gondii*, V775-V816 in *P. falciparum*) was first formed in 1:1.1 molar ratio, respectively, and incubated for 1 h at 4 °C. For measurement, 2 μL of 200 μM TgMLC-S or PfELC were injected 19 times into 20 μM of the pre-formed complex. The measurements were performed with a MicroCal PEAQ-ITC (Malvern) at 25 °C. The data were processed using the MicroCal PEAQ-ITC Analysis Software and fitted with a one-site binding model.

**Bioinformatics methods.** The homologous protein sequences were aligned with the program MAFFT[53]. The protein disorder probability was calculated using the disEMBL[54] server with loops and coils defined by dictionary of secondary structure of proteins[55]. The secondary structure prediction of PfELC, TgELC1 and TgELC2 was calculated in JPred[56].

**Small angle X-ray scattering.** The SAXS data were collected at the P12 BioSAXS beamline[57] at the PETRA III storage ring (DESY, Hamburg, Germany). The concentrated samples of TgELC2 and PfELC (10 mg/mL) were dialyzed against the buffer (20 mM HEPES (pH 7.5), 150 mM NaCl, 0.5 mM TCEP for TgELC2; 20 mM Tris (pH 8.0), 150 mM NaCl, 0.5 mM TCEP for PfELC-N) overnight at 4 °C. Further, the samples were centrifuged (5 min, 15,000 g, 4 °C) and a dilution series of each sample (typically in a range of 0.5–10 mg/mL) and their corresponding solvent were measured at room temperature under continuous flow with a total exposure of 1 s (20 × 50 ms frames). The dimeric complex TgELC2/TgMyoA-C, as well as the trimeric complexes using different constructs, were mixed in 1:1 or 1:1:1 molar ratio, purified by SEC and concentrated to 10 mg/mL prior to the measurement. The X-ray scattering data were measured in an on-line SEC-SAXS mode, using a SD200 Increase column (GE Healthcare) at 0.5 ml/min with 1 frame recorded per second. The sample of PfELC was concentrated to 10 mg/mL and the X-ray scattering was measured in the on-line SEC-SAXS mode, using a SD200 5/150 column at 0.4 mL/min. The automatically processed data were further analyzed using the ATSAS suite[58] programs CHROMIXS[59] and PRIMUS[60] to determine the overall parameters and distance distribution, CRYSOL[61] to compute the scattering from the crystal structures and CORAL[62] to compute the scattering from the crystal structures with dummy residues mimicking the missing flexible parts. The results of all SAXS measurements are summarized in Supplementary Table 3. All SAXS data and models have been deposited in the SASBDB (www.sasbdb.org) with accession codes: SASDH64, SASDH74, SASDH84, SASDH94, SASDHA4, SASDHB4, SASDHC4, SASDHD4 and SASDHE4.

**NMR.** All NMR experiments were conducted on a Bruker Avance II 800 NMR spectrometer equipped with a cryoprobe at 288 K in 50 mM HEPES, 20 mM NaCl, 0.5 mM TCEP and 10% (v/v) $D_2O$ at pH 7.0, except for H(CCO)NH-TOCSY and (H)C(CO)NH-TOCSY experiments that were performed on a Bruker Avance III 600 NMR spectrometer equipped with a room temperature probe. Full-length PfELC (residues 1–134) was $^{15}N$ and $^{15}N^{13}C$ labeled and concentrated to 500 μM. PfELC-N was also $^{15}N$ and $^{15}N^{13}C$ labeled and in addition site-selectively $^{13}C$ labeled[63–65] by using 1-$^{13}C_1$ and 2-$^{13}C_1$ glucose. Samples were concentrated to about 1 mM. All spectra were processed suing NMRPipe[66] and analyzed using NMRView[67].

Backbone resonances of $^{15}N^{13}C$ labeled samples (1–74 and 1–134) were assigned using HNCACB[68] and HN(CO)CACB[69] experiments. Aliphatic side chains (1–74) were assigned using H(CCO)NH-TOCSY[70] (H)C(CO)NH-TOCSY and H(C)CH-TOCSY[71] experiments. Aromatic side chains (1–74) were assigned

by (HB)CB(CGCD)HD[72] and aromatic H(C)CH-TOCSY experiments and verified by the site-selective [13]C labeling.

NOEs for the structure determination were derived from 3D-NOESY-HSQC experiments for [15]N, [13]C aliphatic nuclei and [13]C aromatic nuclei (on 1-[13]$C_1$ and 2-[13]$C_1$ glucose labeled samples). Phi-Psi dihedral angle constraints were derived using TALOS[73]. Structure calculations were performed using ARIA 2.3[74] and standard parameters. The lowest-energy models have been deposited in the PDB with accession number 6tj3. Secondary structure elements were determined from chemical shifts and the dynamics of the PfELC backbone was probed using heteronuclear NOEs ({[1]H}-[15]N NOE). This [15]N based dynamics experiment allowed us to distinguish between rigid ({[1]H}-[15]N NOE > 0.7, secondary structure elements), somewhat flexible ({[1]H}-[15]N NOE ~ 0.5–0.7, loops and turns) and extremely flexible ({1H}-[15]N NOE < 0.5, unfolded/ random coil) regions of the protein. Ramachandran analysis was performed by PROCHECK[75].

{[1]H}-[15]N NOE saturation was performed using a train of shaped 180° pulses in a symmetric fashion[76–78] for 3 s and a total inter-scan relaxation period of 10 s. Data collection, processing and analysis details are summarized in Supplementary Table 4.

**Crystallization**. PfELC-N was concentrated (5 kDa cut-off) to 26 mg/mL and 200 nL of the sample was mixed with 100 nL of reservoir solution (0.1 M Tris-HCl (pH 8.5), 0.2 M $Li_2SO_4$, 30% PEG 4000). The crystals grew in sitting drop plates at 19 °C for 7 days.

The trimeric complex of MLC1-S, TgELC2 and TgMyoA-C (S777-V818) was mixed in a molar ratio of 1.1: 1.1: 1, respectively. After 1 h of incubation, the trimeric complex was separated by gel filtration in 20 mM HEPES pH 7.5, 150 mM NaCl, 0.5 mM TCEP using a Superdex 75 16/600 column (GE Healthcare). The fractions containing the peak of the trimeric complex were concentrated (5 kDa cut-off) to 10 mg/ml. The crystals grew for 7 days at 19 °C in sitting drop plates prepared by mixing 200 nl of the sample with 100 nl of reservoir solution (0.1 M imidazole, 0.1 M MES monohydrate pH 6.5, 20% v/v PEG 500 MME, 10% w/v PEG 20 000, 0.12 M 1,6-hexadiol, 0.12 M 1-butanol, 0.12 M 1,2-propanediol, 0.12 M 2-propanol, 0.12 M 1,4-butanediol, 0.12 M 1,3-propanediol).

The recombinantly expressed dimeric complex of TgELC1 and TgMyoA-C (S777-V818) was mixed with MLC1 in 1:1.1 molar ratio, incubated for 1 h and the trimeric complex was separated by gel filtration in 20 mM HEPES pH 7.5, 150 mM NaCl, 0.5 mM TCEP using a Superdex 75 16/600 column (GE Healthcare). The fractions containing the peak of the trimeric complex were concentrated (5 kDa cut-off) to 10 mg/mL. The crystals of the calcium-bound complex grew within 7 days at 19 °C in a sitting drop prepared by mixing 200 nl of the sample with 100 nL of reservoir solution (20% w/v ethylene glycol, 10% w/v PEG 8000, 0.1 M Tris (base), 0.1 M bicine pH 8.5, 0.09 M sodium nitrate, 0.09 M sodium phosphate dibasic, 0.09 M ammonium sulfate). The crystals of the calcium-free complex grew within 7 days at 19 °C in a sitting drop plate prepared by mixing 200 nL of the sample with 100 nL of reservoir solution (32% w/v PEG 8000, 0.1 M Tris pH 7.0, 0.2 M LiCl).

The recombinantly expressed dimeric complex of MTIP (residues 77–204) and PfMyoA-C (775–816) were mixed with excess of His-tagged PfELC, the complex was purified by NiNTA IMAC, dialyzed and TEV-cleaved overnight at 4 °C and further purified by negative NiNTA IMAC and size exclusion chromatography using Superdex 200 column with 20 mM HEPES pH 7.5, 150 mM NaCl, 0.5 mM TCEP. The crystals grew within 3 days at 4 °C in a sitting drop prepared by mixing 150 nL of the sample with 150 nL of reservoir solution (0.1 M imidazole/MES pH 6.5, 20% w/v ethylene glycol, 10% PEG 8000, 0.03 M of each di-ethylene glycol, tri-ethylene glycol, tetra-ethylene glycol and penta-ethylene glycol).

**Data collection and structure determination**. The diffraction data of the trimeric complexes were collected at the P13 EMBL beamline of the PETRA III storage ring (c/o DESY, Hamburg, Germany) at 0.9762 Å wavelength and 100 K temperature using a Pilatus 6 M detector (DECTRIS). The diffraction data of PfELC-N were collected at the P14 EMBL beamline of the PETRA III storage ring (c/o DESY, Hamburg, Germany) at 1.0332 Å and 100 K temperature using an EIGER 16 M detector (DECTRIS). The diffraction data were processed using XDS[79], merged with Aimless[80] or (STARANISO[81] in case of the *P. falciparum* trimeric complex) and phase information were obtained by molecular replacement with Phaser[82], using the structure of peptide-bound TgMLC1 (PDB ID 5vt9) as a search model in case of the trimeric complexes and the NMR structure as search model in case of PfELC-N. In all cases, the models were further built and refined in several cycles using PHENIX[83], Refmac[84] and Coot[85]. Data collection and refinement statistics are summarized in Table 2. In all structures, over 98% residues are in the favored region of the Ramachandran plot and each structure contains no more than one Ramachandran outlier. PyMOL was used to generate figures, measure the angle of the helical kink, inter-molecular angles, distances and RMSDs. PDBePISA[86] was used to characterize the intermolecular interfaces. The atomic coordinates and the structure factors have been deposited in the PDB with accession numbers 6tj4, 6tj5, 6tj6, 6tj7 and 6zn3.

**Modeling**. The modeling procedure was performed in Modeller version 9.18[87]. We built 50 models for the TgMyoA residues 772–791. These 50 models were fused to

the structure of TgMyoA (PDB ID 6due; residues 33–771). All 50 models were tilting along the bond/dihedral angle between residue 771 and the first modeled residue, that is 772; at the same time, the residues 33–771 of the 6due structure remained fixed. Thus, each of the produced models consisted of an intact crystal structure 6due (till residue 771) and de novo modeled fragment of 772–791. Restraints in a form of i-i + 4 h-bonding pattern were imposed in order to ensure that all 50 models have an α-helical arrangement along the whole length of the de novo modeled fragment, and also at the junction between residues 771 and 772. The crystal structure of complex 1 (PDB ID 6tj5) or complex 2 (PDB ID 6tj7) were superposed on the 50 models using the TgMyoA residues 780–791. After super-position, the modeled conformation of this fragment was removed from the merged structures, which produced models consisting of an intact crystal structure of TgMyoA (PDB 6due), the modelled helix of TgMyoA (residues 772–779) and the intact crystal structure of the complex 1 (50 models) or complex 2 (50 models), starting from the TgMyoA residue 780 of these structures. Next, all reconstructed complexes were screened against the existence of atomic clashes using the Chimera software[88] and the best five models (both complex 1 and complex 2) were energy minimized by executing 1000 steps of conjugate gradient energy minimization in the NAMD program[89]. All energy minimizations were performed in a water box with ions.

**Normal mode analysis**. Normal mode analysis (NMA)[90] was used to probe essential dynamics of the reconstructed trimeric models. The NMA was performed in an all-atom representation on the best five energy-minimized models using the BIO3D software[91]. The deformation analysis was performed, using the first 20, 50 and 100 modes, and also on the first 10 modes separately. This allowed us to not only identify possible hinge points within the studied structures of trimeric complexes, but also to determine which hinges correspond to which modes.

**Statistics and reproducibility**. In all reported experiments, the protein samples were expressed and purified under identical experimental conditions. The figures represent the results from one experiment, unless stated otherwise. The CD experimental curves were recorded 10 times, averaged and buffer-subtracted. The SAXS data recorded in batch mode represent a buffer-subtracted average of 20 measurements of the same sample measured under continuous flow.

**Reporting summary**. Further information on research design is available in the Nature Research Reporting Summary linked to this article.

## Data availability
The datasets generated during and/or analyzed during the current study are available from the corresponding author on request. The data source data underlying the charts in the main and supplementary figures is deposited in Figshare repository[92]. Coordinates and structure factors as well as NMR structures were deposited in the PDB at the Research Collaboratory for Structural Bioinformatics (RCSB) with the following identifying codes: 6tj3, 6tj4, 6j5, 6tj6, 6tj7, 6zn3. The averaged and subtracted SAXS data were deposited in SASBDB with the following identifying codes: SASDH64, SASDH74, SASDH84, SASDH94, SASDHA4, SASDHB4, SASDHC4, SASDHD4 and SASDHE4. The structural models of full lengths MyoA-MLC1-ELCs have been uploaded to Zenodo[93].

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

## Acknowledgements

We thank the Sample Preparation and Characterization facility of EMBL Hamburg for support with nanoDSF, ITC measurements and with protein crystallization. We acknowledge all group members for continuous support and feedback on the project and during manuscript preparation. We would like to thank the group of Thomas R. Schneider at EMBL Hamburg for access to the EMBL beamlines P13 and P14 and Guillaume Pompidor and Grzegorz Chojnowski for help with data processing and initial model building. This work was funded by a joint grant from the Joachim Herz Foundation (Grant number: 800026) to Tim Gilberger and to Christian Löw. Open Access funding enabled and organized by Projekt DEAL.

## Author contributions

Designed research: S.P., K.K., J.K., U.W. and C.L.; Performed research: S.P., K.D., K.K., H.M., U.W., C.L.; Analyzed data: S.P., K.D., K.K., H.M., T.G., D.S., J.K., U.W., C.L.; Prepared figures: S.P.; Wrote the paper: S.P. and C.L. and all other authors contributed to writing of the manuscript.

## Funding

## Competing interests

The authors declare no competing interests.
