## [Peer Review File · Communications Biology]

Reviewers' comments:

Reviewer #1 (Remarks to the Author):

Summary

The glideosome is a molecular complex that is specific to the apicomplexan parasites and essential for their virulence and survival. Although the components of the complex are known and conserved across the phylum, their interaction between each other as well as the immobilization of the complex within the pellicle remain largely unknown.

In this study, Pazicky and colleagues describe in details the structure of the last component identified, the essential light chains of *Toxoplasma gondii* (TgELC1-2) and *Plasmodium falciparum* (PfELC), alone or in the context of the neck region of the associated motor (MyoA) and its light chain TgMLC1/PfMTIP.

The study is well conducted. The structure of the “trimeric” complex confirms the observation made previously by ITC and HDX and gives the molecular detail of the interaction of TgMyoA-neck with its light chains. In addition, the reported folding of the TgMyoA-neck induced by the binding of TgMLC1 and/or TgELC likely explains the *in vivo* observed data where TgMyoA and PfMyoA were totally destabilized upon deletion of TgMLC1 or TgELCs or PfMTIP (Egarter et al., 2014; Williams et al., 2015; Sebastian et al., 2012). This structure also confirms at the molecular detail how this atypical short myosin can function as a conventional myosin generating force rather than movement. It is therefore of special interest beyond the scope of parasitology.

Specific comments:

1. From the crystal structure of the *T. gondii* “trimeric” complex, is it possible to establish a model of the *Plasmodium* “trimeric” complex to compare the interaction taking place between the 3 components, knowing that the structure of PfMyoA-neck-PfMTIP has been determined?
2. Fig1a, the scheme of the glideosome needs to be revised as well as its description in the introduction, see for example Bookwalter et al., JBC, 2014 or Powell et al., JBC, 2019 for a scheme taking account of the known topology information.

More specifically:

- GAP45 is the only protein of the complex to be myristoylated and palmitoylated and these N-terminal acylation anchor the protein to the plasma membrane and not to the IMC. It is interacting with the IMC by its C-terminus. This is true in both *Toxoplasma gondii* (Fréchal et al., 2010) and in *Plasmodium falciparum* (Ridzuan et al., 2012).
- MLC1/MTIP is not predicted to be myristoylated. It is likely palmitoylated on C8-C11 but its interaction with the IMC could also be via protein-protein interaction (Fréchal et al., 2010).
- the topology of GAP50 is inverted, most of the protein is in the IMC lumen and only 6 amino acids are supposed to interact with the rest of the complex.

Minor comment

Line 255: “W779-F77” should be “W779-F79”

Reviewer #2 (Remarks to the Author):

The manuscript describes the structure of a trimeric complex consisting of myosin A, essential light chain and myosin light chain. This complex plays a role in the gliding motility of apicomplexan parasites. In addition to crystallographic data, thermodynamic and kinetic data are presented. The manuscript is interesting, providing additional insights into an important virulence machinery. However, several issues require the authors' attention.

1. The text is verbose and a grinding read. Particularly, the results would benefit from shortening, e.g., by removing all discussion.

2. The manuscript appears overloaded with display items, which, in turn, dilutes the key findings. Therefore, one should consider moving some of the tables and subfigures to the supplementary information. In particular, Tables 1 and 2, Figure 1B and C and possibly Figure 2 seem to be dispensable.

3. My main concern is the lack of functional data in support of the 3D structure and, hence, of the conclusions drawn from the structures with regard to parasite biology and virulence. Particularly, residues involved in myosin A regulation, such as S102 TgELC2 or the Ca²⁺ binding EF hands, are attractive candidates for functional studies. Since both *T. gondii* and *P. falciparum* are readily amenable to genetic manipulation, it should be possible to generate the appropriate mutants and discuss the resulting phenotypes in the context of the structural model proposed by the authors.

Point-by-point response to the referees' comments on the manuscript entitled: "Structural role of essential light chains in the glideosome assembly"

General response: We would like to thank the reviewers for their suggestions and comments. We have revised the manuscript, restructured the text to improve readability and flow and included additional experimental data to address the major concerns.

Reviewer #1 (Remarks to the Author):

Summary

The glideosome is a molecular complex that is specific to the apicomplexan parasites and essential for their virulence and survival. Although the components of the complex are known and conserved across the phylum, their interaction between each other as well as the immobilization of the complex within the pellicle remain largely unknown.

In this study, Pazicky and colleagues describe in details the structure of the last component identified, the essential light chains of *Toxoplasma gondii* (TgELC1-2) and *Plasmodium falciparum* (PfELC), alone or in the context of the neck region of the associated motor (MyoA) and its light chain TgMLC1/PfMTIP.

The study is well conducted. The structure of the "trimeric" complex confirms the observation made previously by ITC and HDX and gives the molecular detail of the interaction of TgMyoA-neck with its light chains. In addition, the reported folding of the TgMyoA-neck induced by the binding of TgMLC1 and/or TgELC likely explains the in vivo observed data where TgMyoA and PfMyoA were totally destabilized upon deletion of TgMLC1 or TgELCs or PfMTIP (Egarter et al., 2014; Williams et al., 2015; Sebastian et al., 2012). This structure also confirms at the molecular detail how this atypical short myosin can function as a conventional myosin generating force rather than movement. It is therefore of special interest beyond the scope of parasitology.

Response: We would like to thank reviewer #1 for the positive summary and considering our research well conducted and of general interest. We have addressed specific comments below.

Specific comments:

1. From the crystal structure of the *T. gondii* "trimeric" complex, is it possible to establish a model of the *Plasmodium* "trimeric" complex to compare the interaction taking place between the 3 components, knowing that the structure of PfMyoA-neck-PfMTIP has been determined?

Response: To investigate this highly interesting comparative question we used the last months to crystallize and determined also the structure of the *P. falciparum* trimeric complex. This allows now the direct structural assessment of these homologue structures within related parasite species. For instance, it shows that binding of PfELC is structurally distinct from TgELCs. We have added this novel data to the current manuscript and present the additional crystal structure in Fig. 3g-h.

2. Fig1a, the scheme of the glideosome needs to be revised as well as its description in the introduction, see for example Bookwalter et al., JBC, 2014 or Powell et al., JBC, 2019 for a scheme taking account of the known topology information.

More specifically:

- GAP45 is the only protein of the complex to be myristoylated and palmitoylated and these N-terminal acylation anchor the protein to the plasma membrane and not to the IMC. It is interacting with the IMC by its C-terminus. This is true in both *Toxoplasma gondii* (Fréchal et al., 2010) and in *Plasmodium falciparum* (Ridzuan et al., 2012).

Response: We have updated Fig. 1 according to the suggested literature. In the current figure, GAP45 stretches between the plasma membrane and the IMC and both hypotheses to anchor GAP45 in the IMC are represented: (a) interaction with GAP50 and/or (b) via acylation at its C-terminus.

- MLC1/MTIP is not predicted to be myristoylated. It is likely palmitoylated on C8-C11 but its interaction with the IMC could also be via protein-protein interaction (Fréchal et al., 2010).

Response: We have corrected this error (it is not predicted to be myristoylated, line 73 in Introduction). At the same time, we leave the interpretation of the figure to the reader, as both IMC-anchoring options (N-terminal palmitoylation, protein-protein interaction or both) are open.

- the topology of GAP50 is inverted, most of the protein is in the IMC lumen and only 6 amino acids are supposed to interact with the rest of the complex.

Response: We have reviewed the literature about GAP50 and corrected the figure concerning the GAP50 orientation in the IMC membrane in Fig. 1.

Minor comment

Line 255: “W779-F77” should be “W779-F79”

Response: We have corrected this error in the manuscript (lines 185 and 189).

Reviewer #2 (Remarks to the Author):

The manuscript describes the structure of a trimeric complex consisting of myosin A, essential light chain and myosin light chain. This complex plays a role in the gliding motility of apicomplexan parasites. In addition to crystallographic data, thermodynamic and kinetic data are presented. The manuscript is interesting, providing additional insights into an important virulence machinery. However, several issues require the authors' attention.

Response: We would like to thank the reviewer #2 for the evaluation of our manuscript. We addressed the raised points below.

1. The text is verbose and a grinding read. Particularly, the results would benefit from shortening, e.g., by removing all discussion.

Response: We have restructured the main text of the manuscript and moved technical details to the Methods section and Figure descriptions. With this change, we believe that the manuscript is easier to read and the flow of the text has improved. As suggested, we have also removed the discussion section and combined relevant discussion points with the results part.

2. The manuscript appears overloaded with display items, which, in turn, dilutes the key findings. Therefore, one should consider moving some of the tables and subfigures to the supplementary information. In particular, Tables 1 and 2, Figure 1B and C and possibly Figure 2 seem to be dispensable.

Response: In the context of rewriting the manuscript we have also reorganized the figures, moved several display items to supplementary information as suggested and also moved several tables to the supplementary information section.

3. My main concern is the lack of functional data in support of the 3D structure and, hence, of the conclusions drawn from the structures with regard to parasite biology and virulence. Particularly, residues involved in myosin A regulation, such as S102 TgELC2 or the Ca²⁺ binding EF hands, are attractive candidates for functional studies. Since both *T. gondii* and *P. falciparum* are readily amenable to genetic manipulation, it should be possible to generate the appropriate mutants and discuss the resulting phenotypes in the context of the structural model proposed by the authors.

Response: Following the reviewer's recommendation, we have performed additional *in vitro* experiments to study the role of Ca²⁺ in regulating complex assembly as well as the role of ELC phosphorylation. To address the role of Ca²⁺, TgELC1 and TgELC2 were refolded in the presence of 5 mM Ca²⁺ or 5 mM EDTA to ensure calcium saturation or elimination. ITC measurements revealed that in both cases, the affinity drops only twofold in the absence of calcium, which is a much smaller effect than previously reported by Powell et al (2018). But our results are supported by the calcium-free structure, which only differs marginally from the calcium bound form. Therefore, we believe that the mutational analysis *in vivo* would not result a significant phenotype but we rather speculate that calcium ions cause a secondary effect e.g. by regulating different kinases which in turn phosphorylate glideosome members. We have added and discussed these new results in lines 285-304, Fig. 6c, Supplementary Fig. 5c and Table 1.

To address the role of phosphorylation in ELCs, we performed phosphomimetic mutations of residues that were shown to be phosphorylated *in vivo*: TgELC2 (S102) and PfELC (S127). In both cases, we observed a decrease of the affinity of ELCs to the respective MyoA peptides. We added these results in the manuscript: lines 191-194, Table 1 and Supplementary Fig. 3b for TgELC2-S102 and lines 217-226, Table 1 and Supplementary Fig. 3d for PfELC-S127.

We agree that it highly attractive to generate the appropriate mutants in *T. gondii* and *P. falciparum* and analyse the resulting phenotypes in the context of our structural model, but this is beyond the scope of this manuscript. The generation of mutant parasites in both parasite species is still very time consuming specially if conditional systems are advisable.

REVIEWERS' COMMENTS:

Reviewer #1 (Remarks to the Author):

The authors have extensively revised the text and the manuscript is now easier to read. They addressed all my main concerns and even provided an additional structure. This manuscript is of interest and timely knowing that several other structures of the complex will be published soon (at least two manuscripts are available in BioRxiv since a few days).

Reviewer #3 (Remarks to the Author):

The authors have substantially improved the manuscript. It is now an exciting read, and the amount of data presented is nothing but impressive.

I have only one minor suggestion with regard to the end of the text, which appears to fade out instead of ending on a high note. Since the authors have decided to combine the results and the discussion, it is recommended to end the text with a one-paragraph conclusion.